



# A geological model for the management of subsurface data in the urban environment of Barcelona city.

Enric Vázquez-Suñé[1,2], Miguel Ángel Marazuela[1,2,3], Violeta Velasco[1,2], Marc Diviu[1,2], Andrés Pérez-Estaún[4], Joaquina Álvarez-Marrón[4]

[1] Institute of Environmental Assessment and Water Research IDÆA - CSIC. Jordi Girona 18, 08034 Barcelona, Spain
    [2] Associated Unit: Hydrogeology Group (UPC-CSIC)
    [3] Department of Geotechnical Engineering and Geosciences, Universitat Politècnica de Catalunya (UPC), Jordi Girona 1-3, 08034 Barcelona, Spain
    [4] Institute of Earth Sciences Jaume Almera ICTJA - CSIC, Lluís Solé Sabarís s/n, 08028 Barcelona, Spain

Correspondence to: Enric Vázquez-Suñé  (enric.vazquez@idaea.csic.es)

**Abstract.** The overdevelopment of cities since the industrial revolution has shown the need to incorporate a sound geological knowledge in the management of required subsurface infrastructures and in the assessment of increasingly needed groundwater resources. Also, the scarcity of outcrops and the technical difficulty to conduct underground exploration in
urban areas highlights the importance of implementing efficient management plans that deal with the legacy of heterogeneous subsurface information. To deal with these difficulties, a methodology has been proposed to integrate all the available spatiotemporal data into a comprehensive spatial database and a set of tools that facilitates the analysis and processing of the existing and newly added data for the city of Barcelona (NE Spain). Here we present the resulting actual subsurface 3D geological model that incorporates and articulates all the information stored in the database. The methodology
applied to Barcelona city benefited from a good collaboration between administrative bodies and researchers that enabled the realization of a comprehensive geological database despite logistic difficulties. Currently, the public administration and also private sectors, both benefit from the geological understanding acquired in the Barcelona city. For example, when preparing the hydrogeological models used in groundwater assessment plans. The methodology further facilitates the continuous incorporation of new data for the implementation and sustainable management of urban groundwater and also contributes to
significantly reduce the costs of new infrastructures.

## 1. INTRODUCTION

The need for a groundwater supply has led people to settle near natural water sources since ancient times, usually near major rivers. For this reason, many of the world's major cities are located above alluvial or deltaic aquifers (Carlson et al., 2011; Davies, 2015; Gleeson et al., 2015; Tessler et al., 2015). Currently, particularly with the industrialization that began in the
1950s, most of these aquifers have been severely affected in terms of the quantity and quality of their waters (Aeschbach-Hertig and Gleeson, 2012; Gleeson et al., 2012b).



Urban expansion generates an enormous demand for infrastructure and, paradoxically, progressively less space for construction (Grimm et al., 2008). As a result, cities must grow downward, which requires subterranean construction (e.g., transportation networks and underground parking garages). This subterranean infrastructures usually interact with shallow aquifers producing frequently negative repercussion on them (Attard et al., 2015; Ferguson, 2004; Ferguson and Woodbury,

2007; Kazemi, 2011; McDonald et al., 2014; Menberg et al., 2013; Taniguchi et al., 2008; Vázquez-Suñé et al., 2005; Vörösmarty et al., 2000). Various contamination pathways associated with this problem include spillways, sewage networks, and discharges into wells. Another result of urbanization is an abundance of hardscapes, which severely reduce the amount of aquifer recharge (Vázquez-Suñé et al., 2010).

The European Water Framework Directive was developed in response to these problems faced by society in the 21st century

(WFD, 2000). Its objective was to unify water management in the European Union by defining long-term needs and actions to be performed in various countries. It is an important step in the management and assessment of groundwater in general. However, its norms are typically too lax and do not provide for adequate management of urban aquifers required for the sustainable, controlled exploitation of water and geothermal resources (Epting et al., 2013; García-Gil et al., 2015b; Gleeson et al., 2012a; Vázquez-Suñé et al., 2006).

The literature contains several solutions and ideas for general groundwater management or for management of specific problems in various countries. (García-Gil et al., 2015c; Parriaux et al., 2004; Zhou et al., 2015). Almost all of these studies conclude that numerical models of current and future aquifer conditions are the primary tool for accomplishing this objective. However, the development of comprehensive numerical models requires 3D geological models that are difficult to construct in an urban environment (Huggenberger and Epting, 2011).

To overcome these difficulties, extensive geological knowledge of the subsurface is crucial for construction planning and management of groundwater and for security reasons (Huggenberger et al., 2013; Santanach et al., 2011; Velasco et al., 2012). However, geologic features are increasingly difficult to observe due to growing urbanization, and the difficulty of performing geological and geophysical exploration in dense urban areas leads to a need for specific exploratory techniques.

In cities, it is very important to compile existing historical information. Although a large amount of information is typically

available, much of this information is not stored in standardized or ordered formats but rather is dispersed among the various companies and public institutions that generated them using different protocols and criteria. Close collaboration is needed between an administration supplying the data and a research team investigating the geology (de Rienzo et al., 2008; Di Salvo et al., 2012; Velasco et al., 2013). New methods of managing geologic data in urban areas are necessary: without these, it is much more complex and difficult to develop management plans for aquifers and subterranean construction in densely

populated environments.

Thus, the objectives of this study are (1) to develop a method for integrating geologic, hydrogeological, and geophysical data from an urban environment into a standardized, accessible database using effective data-management tools and (2) to show how this challenge was met in the city of Barcelona, i.e., how the geology was modelled and consequently how better management of the aquifer was achieved.



## 2. REGIONAL SETTING

The study area is located along the northeastern coast of the Iberian Peninsula and includes much of the cities of Barcelona, Sant Adrià, Badalona, and Santa Coloma de Gramenet (Spain) (Figure 1). It is bounded on the west and northwest by the Collserola Range and on the east and southeast by the Mediterranean Sea.

The topography of the municipalities of Barcelona can be divided into three geomorphic areas. First, the Collserola and Marina Ranges trend NE-SW parallel to the coast and consist primarily of Paleozoic granitic and metamorphic rocks. Second, Paleogene and Neogene rocks are exposed in a band parallel to these ranges and along a fault; these rocks are mostly covered by alluvial deposits eroded from the steep adjacent slopes. Third, underlying gently sloping plains extending to the coast, there are Quaternary detrital materials associated with the deltas of the Llobregat and Besós Rivers and the

coastal plain of Badalona. Most of the study area, particularly the lowlying portion, has been strongly modified by anthropic activity, and many exposures of the geology have been covered or destroyed. Most of the studied area is located on the littoral plain of the Catalan Coastal Ranges and the Besòs River delta.

The Catalan Coastal Ranges are the western onshore margin of the Valencia trough, which has a NE-SW orientation controlled by a set of faults that dip SE and have experienced repeated offsets throughout their history. This fault system is

200 km wide and 400 km long (Banda and Santanach, 1992). These extensional faults have been active since the late Oligocene, primarily during the Miocene, and are still active today (Roca and Guimerà, 1992).

The geometry and depositional characteristics of the Paleogene and Neogene deposits display evidence of two major episodes of deformation. The first episode was one of compressive intraplate tectonics associated with the Alpine orogeny and was recorded in lower Oligocene deposits in an offshore piggyback basin. Simultaneously, the foreland basin of the

Pyrenees (Ebro Basin) subsided. The second episode was extensional and included syn-rift and post-rift stages. The syn-rift stage was recorded in upper Oligocene and lower Miocene deposits that filled a graben created during this period. The post-rift stage began in the late Langhian and extends to the present. It is characterized by an attenuation of tectonic activity and significant deposition that covered the horsts developed in the previous episode (Roca et al., 1999; Salvany, 2013). Syntectonic erosion of compressive structures generated during the Paleogene and Neogene exhumation associated with the

extensional phase altered the amplitudes and ratios of vertical movements (Gaspar-Escribano et al., 2004; Lewis et al., 2000). This erosion exposed Variscan metamorphic and intrusive rocks in the littoral and prelittoral chains (Solé et al., 2002). The plain of Barcelona is underlain primarily by alluvial deposits eroded from the Collserola Range. Its northeastern portion is underlain by Holocene deposits of the Besós River.

## 3. MATERIALS AND METHODS

This study of the geology of Barcelona reflected the particular local urban characteristics. The approach involved collecting historical data and acquiring new data.



### 3.1. Previous data and historical studies

Collecting historical data in an urban environment is important yet complex due to the difficulty of obtaining new data and the inability to observe exposures destroyed in recent decades by heavy urbanization. Of all the material consulted, technical reports prepared during the 20th and early 21st centuries stand out due to the large amount of subsurface information they

contain. In some cases, these reports contain the only records of ancient exposures.

Certain researchers (e.g., Llobet-Vall-llosera, 1840) performed morphological and hydrogeological studies of the city of Barcelona, which included the network of important streams in Barcelona. They also noted the major lithologic characteristics of the materials underlying the region based on findings from 34 wells. In 1891, a topographical and hydrogeological study of the city provided a list of 27 wells with lithologic descriptions (García-Faria, 1891). Almera (1894)

studied the Pliocene materials. The first study of the Besós River delta was performed in the late 19th century and yielded the first data regarding its composition (Moracas, 1896). In 1900, the Council of Barcelona published the first geological map of Barcelona at a scale of 1:40,000. These studies provide an understanding of the original stream network, much of which has been channelized and diverted.

At the beginning of the 20th century with the study by Rubio and Kindelan (1909), which was based on data from numerous

wells (some as deep as 173 m), there was a substantial improvement in the understanding of the subsurface below Barcelona. The map of Barcelona based on the American flight of 1956 at a scale of 1:50,000 is highly useful in mapping local ancient geologic exposures. In the middle of the 20th century, the first study of the Quaternary deposits was performed (Solé-Sabarís, 1963).

Geological mapping at a 1:50,000 scale from the second MAGNA series of IGME is available (Alonso et al., 1977). This

work was later integrated with the current ICGC (ICGC, 2005) and with the 1:25,000-scale geological map by Ventayol et al. (1978), Ventayol (2016). Sanz Parera (1988) compiled and synthesized geologic and hydrogeologic information regarding the Barcelona Plain; this information was updated by Riba and Colombo (2009).

### 3.2. New data acquisition and methodology

The absence of outcrops and the difficulty of applying certain techniques often used in geological exploration have meant

adapting these techniques or, in many cases, developing new techniques to achieve the objectives.

During the previous 20 years, the Hydrogeology Group (CSIC-UPC) has performed extensive hydrogeological research in the vicinity of Barcelona, prompted mostly by the local governments of Barcelona, Santa Coloma, Badalona, etc. These studies focused on the management of groundwater resources and on the hydrogeological aspects of excavations, tunnels (e.g., the Metro and High Speed Train), drainages, and other infrastructures. These studies generated a large amount of

geologic and hydrogeological information that improved our knowledge of the subsurface below Barcelona.

The major source of information for this study consisted of exploratory surveys. First, a review of all existing surveys was performed, and all that were relevant were reinterpreted. Second, newer surveys that were part of various projects in which



the research group was involved were incorporated. This review led to a total of 714 surveys currently available in the study area, or 1462 surveys including those that were performed in the immediate vicinity. Furthermore, additional observations consisted of detailed studies of the limited outcrops available in the city parks and the road cuts as well as geologic interpretation of old aerial photographs (taken in the 1950s just before urban sprawl affected the area).

All the information from construction projects, surveys, and nearby exposures together with geophysical profiles (Martí et al., 2008; Gámez et al., 2009; Velasco, 2013; Velasco et al., 2012a and 2012b) were standardized and entered into the geodatabase for modelling and analysis in a Geographic Information System (GIS) environment; concretely in ArcGIS (ESRI). All this enabled us to build a 3D geological model following the methodological approach represented in Figure 2. Some of these results are described in the following sections.

**4. GEOLOGICAL MODEL**

The geological model presented in this paper, include the entire stratigraphic succession ranking from Palaeozoic materials at the base, to the present top surface.

The Quaternary deposits unconformably overlie a heterogeneous substrate of various ages. First, a synthesis of the lithologic and structural characteristics of the pre-Quaternary substrate are described (Figures 3); these consist of Paleozoic, Triassic,
Miocene, and Pliocene materials. Later, the Quaternary stratigraphy of the Besós River delta is described (Figures 4 and 5).

**4.1. Paleozoic rocks and structural evolution**

The Paleozoic substrate consists of granitic and metamorphic rocks (Figure 3). The granitic rocks belong to the Paleozoic batholith that forms the core of the Catalan Coastal Range and the basement of the coastal shelf. They are plutonic rocks of granodioritic composition emplaced during the Carboniferous/Permian (Solé et al., 2002). During the latest stages of
emplacement a large networks of porphyry dikes (locally granodioritic to granitic) and aplites, pegmatites, and porphyritic leucogranites of centimeter to meter scale were intruded. Physicochemical alteration of the granodiorites along joints resulted in scattered sandy material with variable amounts of clays and iron oxyhydroxides, locally referred to as the lehm. It is also often present in caliche horizons associated with major alteration zones, particularly in the lowlying areas, where the granites are more mafic.
The Paleozoic metamorphic rocks, are Ordovician shales, schists, and quartzites and Devonian limestones. Their northwestern margin is defined by a normal fault. Two distinct, laterally extensive series can be distinguished. These rocks were intensely folded during the Hercynian orogeny and were affected by the intrusion by the granitic batholith in the form of a distinct zone of contact metamorphism. This metamorphism decreases with distance from the intrusion and created three primary types of metamorphic rocks: hornfels, pelitics in the contact zone, high-grade calc-silicate schists formed from a
carbonate protolith, and schists, which represent a transition to nonmetamorphosed phyllites (Sanz Parera, 1988). During the Alpine orogeny, these rocks were not easily folded, which resulted in brittle fracturing and a dense network of faults. These





faults separate various blocks between the Collserola Range and the sea. In addition to these major faults, there are numerous joints, fractures, and local tectonic deformations. Additional evidence of intense tectonic activity is provided by the presence of mylonite.

### 4.2 Triassic—Neogene units and structural evolution

The Mesozoic rocks, specifically those of the Triassic, are limited to the hill of Montgat, which is isolated by a series of NE-SW-trending fractures. These rocks consist of conglomerates interbedded with thick reddish clayey siltstones belonging to the Buntsandstein facies. There are also limestones and dolomitic limestones of the lower Muschelkalk unit. Locally, they are positioned between two dolomitic beds, i.e., red claystones with evaporitic deposits that can be assigned to the Muschelkalk or to the Keuper. This entire sequence displays contacts that are highly fractured, which hinders accurate

identification of the various rock types.

The oldest Cenozoic materials in the study area are Miocene and are restricted to the area of Badalona. They have an estimated thickness of 150 m and form small hills. They consist of breccias and generally massive quartzite sandstones that are reddish gray at the base and among which are sandwiched a complex of much less competent sandstones to claystones. The top of the series, which crowns the hills, consists of alternating yellow claystones, sandstones, and small conglomerate

lenses. These rocks constitute another of the laterally extensive series that underlie the region and whose margins contain scattered cataclastites.

The youngest Cenozoic deposits are Pliocene and consist primarily of two distinct units. The lower unit consists of clay and blue marl of marine origin. The upper unit consists of sands and gravels with a sandy clay matrix that were derived from the coastal range. The Pliocene deposits vary greatly in thickness because their base conformed to the paleotopography, which

was controlled by staggered fault blocks extending from the mountains to the sea. These deposits are sandier toward the interior and loamier toward the coast and were incised by numerous streams.

All these geologic units constitute the substrate of the Quaternary formations. The geometry of this substrate resulted from the complex local tectonism alternating with periods of severe erosion (Figure 3). Generally, this paleotopography has a funnel morphology that rapidly opens to the southeast. The slope gradient is steeper near the Collserola Range and gentler

near the sea. The paleotopography was affected by gully incision and by faulting. In many locations, the structure strongly controlled the drainage network development, the clearest example being the Besós River, which largely coincides with one of the major NW-SE-trending faults. In addition, NE-SW-oriented faults also control the paleotopography developed on the pre-Quaternary units in the form of parallel steps extending to the coast.

### 4.3. Quaternary Stratigraphy

Overlying the irregular pre-Quaternary topographic surface are Quaternary formations with a maximum total thickness of 30 meters (Figure 4, Figure 5). The Pleistocene deposits consist of ancient alluvial fans. Their base nearest the Collserola Range is characterized by subangular gravel in a red clay matrix. This basal interval grades seaward into sands and reddish clays





and finally to red clays nearest to the sea. The Pleistocene deposits include a layer of brown and yellow eolian silt with numerous carbonate nodules (locally known as *cervell de gat*). A weathering profile dominated by a carbonate-rich crust (locally known as *tortorà*) is developed locally to a thickness of 1 m on the eolian silt. The alternation of these intervals is known locally as *tricicle*.

The Holocene deposits are the most extensive and consist of alluvial deposits and the Besós River delta complex. The first unit consists of fine detrital materials (reddish clay and silt) with carbonate nodules. This unit is homogeneous, has a debris-flow or alluvial origin and includes alluvial fan deposits located at the foot of the mountain range. The next unit consists of stream-channel gravel and sand derived from the weathering of granites from the mountain range. The last unit consists of alluvial deposits strongly modified by anthropic processes. Finally, covering much of the study area is the Besós River delta

complex.

The Besós River delta complex is divided into two sequences with thicknesses on the order of tens of meters and bounded on the bottom and top by erosional surfaces. These surfaces represent periods of subaerial exposure probably due to Quaternary glacial-eustatic sea-level changes. The lower sequence consists primarily of distinct continental sediments and fluvial channel and floodplain facies. The upper sequence records deposition in various environments ranging from subaerial to

marine. This sequence contains three systems tracts: a lower late lowstand systems tract, a transgressive systems tract, and an upper regressive systems tract (Velasco et al., 2012a). In total, there are six distinct stratigraphic units within the Besós River delta complex, which are designated units A through F.

Unit A is the lowermost in the lower sequence and consists primarily of poorly sorted gravel deposited on the pre-Quaternary substrate. Lenticular geometries are observed in transverse sections, and tabular geometries are observed in

longitudinal sections. The distribution of this unit is limited to the lowest areas of the paleotopography. The origin of these deposits is linked to relative sea-level fall during the lowstand, at which time the river channels shaped the paleotopography.

Unit B unconformably overlies unit A or the pre-Quaternary units and is laterally extensive. This association of facies consists primarily of gravel with mud lenses that grade laterally to muddier intervals. This unit is interpreted as a facies deposited by high-sinuosity channels during a period of relative sea level rise and transgression.

Structurally, unit A is typically affected throughout by existing tectonic structures, whereas unit B typically represents the final stage of deformation and displays no sedimentary deformation.

Unit C1 unconformably overlies unit B, unit A, or the pre-Quaternary units. It consists of bodies of poorly sorted gravel that are concave upward in transverse cross sections and tabular in longitudinal sections.

Unit C2 consists of lenses of well-sorted gravel that resulted from reworking of material from unit C1. The contact between

units C1 and C2 is interpreted as a transgressive surface, and the upper contact bounding unit C2 represents the maximum flood surface.

Unit D consists of very thick beds of clay and gray silt with minor lenses of sand and contains marine fossils. Wedge-shaped geometries are observed in longitudinal sections, and lenticular geometries are observed in transverse cross sections. Based





on its characteristics, this progradational unit was deposited in the pro-delta area during a sea level rise during the highstand stage.

As is unit D, unit E is restricted to the coastal zone. It consists of sand and gravel lenses with a certain fraction of fine material and traces of marine fossils. It corresponds to the progradational facies of the delta front.

At the top of the stratigraphic series is unit F. It consists of lenses of sand, gravel, and mud interbedded with thick mud and conglomeratic packages. These deposits are assigned to channel-fill, overbank, and alluvial subfacies that are collectively interpreted as a proximal floodplain facies.

The youngest unit consists of localized anthropogenic fill deposits. It is heterogeneous in composition and varies widely in thickness.

**4. DISCUSION**

Barcelona provides a good example of sweeping changes in the management of water resources and the importance of collaboration between government and researchers in the advancement of geological knowledge. The Hydrogeology Group (GHS) has spent many years working with the city administration on various projects, such as the project being performed with Barcelona Cicle de l'Aigua S.A. (BCASA) to create an efficient system of storing, displaying, and analyzing all

geohydrochemical data collected in the city.

Barcelona is located in a coastal environment, which lends it particular hydrogeologic characteristics and a particular vulnerability to anthropic processes. Since the Industrial Revolution, many hydrogeologic problems have affected this urban area, such as, primarily fluctuations in groundwater levels, contamination of groundwater, and interactions between infrastructures and the aquifers (Vázquez-Suñé et al., 2010; Vázquez-Suñé et al., 2005).

Since the mid-19th century, particularly in the 21st, aquifers below the city have been heavily exploited. This sustained exploitation produced a sharp drop in groundwater levels. In some areas, the levels dropped to below sea level, which led to salinization and a loss of water quality. However, since the 1970s, an economic crisis and pressure from urban residents caused the disappearance of many industries or their relocation to industrial parks on the outskirts of the city. These changes caused a decrease in groundwater extraction and a recovery of groundwater levels. This new situation led to flooding of a

great deal of existing underground infrastructure (Ondiviela-Monté et al., 2005; Vázquez-Suñé et al., 2003).

The industrialization is also one of the sources of aquifer contamination in the area. A great deal of industrial waste was discharged directly into wells or was buried, thereby contaminating the soil. Today, these historical practices and discharges from the wastewater treatment plant (WWTP) and sewers are one of the main causes of groundwater contamination (Jurado et al., 2014a; Jurado et al., 2014b; Jurado et al., 2012b; Tubau et al., 2010).

Another major problem that the city faces is the interactions between underground infrastructures and aquifers. The dense network of underground tunnels in Barcelona (High Speed Train, Metro, etc.) could creates barriers to groundwater flow and causes a rise in groundwater levels upstream of the structure and a drop downstream (Font-Capo et al., 2015; Font-Capó et



al., 2011; Pujades et al., 2012; Pujades et al., 2015). Investigations for these types of construction projects often contribute to the local body of subsurface information (Filbà et al., 2016; Font-Capo et al., 2012; Martí et al., 2008). In addition, excavations in the urban environment have required special measures to decrease groundwater levels temporarily and allow for proper performance of the project (Jurado et al., 2012a; Pujades et al., 2014a; Pujades et al., 2014b). In recent years, there

has been strong interest in developing geothermal energy (Lund and Boyd, 2016), and various projects for generating energy from groundwater are being constructed in Barcelona and Santa Coloma de Gramenet.

In recent decades, municipalities and public administrations (municipalities of Barcelona, Badalona and Santa Coloma de Gramenet and the Catalan Water Agency –ACA-), as well as some private companies, have boosted research in this area after becoming aware of the importance of geology in groundwater management and the value of compiling and conserving

relevant existing information and new information being generated.

For the management of groundwater in Barcelona, the Hydrogeology Group (GHS) proposes a system (Figure 2) whose core is a standardized, geodatabase containing all existing data and to which new data can be continuously added (Velasco, 2013). A set of tools developed for ArcGIS are coupled to this geodatabase and facilitates the visualization, processing, and analysis of the data. Four primary tools were developed for this purpose. HEROS was developed for analyzing, interpreting

geological data for the generation of 3D geological models. (Velasco et al., 2012b). HEROS 3D was designed to improve the 3D visualization and the modelling of 3D surfaces (Alcaraz, 2016). QUIMET permits the visualization and analysis of hydrogeochemical data (Velasco et al., 2014). HYYH was developed for the analysis and interpretation of further hydrogeological data such as heads, abstractions or aquifer tests (Criollo et al., 2015).

These efforts have resulted in an accurate 3D model (Figure 6) of the stratigraphy and structure beneath Barcelona. Because

this model is connected to the geodatabase, which contains all available geologic, hydrogeologic, and geochemical data collected in the city, the data can be rapidly accessed and used at any time.

The 3D geological model is also the basis of the numerical model of the aquifer behaviour and for developing an action plan for sustainable exploitation of the groundwater and its energy resources. Two examples of this process at work in the management of geothermal resources in the vicinity of Barcelona are the studies by Alcaraz et al. (2016) and García-Gil et

al. (2015a).

As a result, quality geologic information can be obtained in an urban environment where exposures are scarce and geophysical surveys are difficult to perform. From an administrative standpoint, this thorough knowledge of the geology of the city acquired by researchers yields a 3D geological model that, coupled with GIS tools developed by the research group, will allow for effective management of the groundwater and energy resources below Barcelona. The administration and

private companies then feedback information that contributes to further research.





## 5. CONCLUSIONS

This study shows the need for a symbiotic relationship between government and research groups for efficient management of geologic data in urban environments. The strategy for achieving this goal requires that government agencies first make available existing geologic, geotechnical and hydrologic data and those tools are then developed to manage the existing data
and any new data that are acquired. This strategy is being applied in Barcelona, where a standardized geodatabase has been developed to manage the available data and incorporate new data rapidly and efficiently. We have developed a set of tools within a GIS environment that facilitate the representation and processing of 3D data. By connecting the 3D geological model with the geodatabase, it is possible to start with a 3D model and create a powerful tool for managing aquifers and their interactions with construction projects. Thus, despite the difficulty of investigating the subsurface in an urban environment, a
highly detailed geological model has been made available.

Finally, through its implementation, both the city administration and private companies benefit from the feedback of geologic knowledge acquired during this process, thereby substantially reducing the cost of construction projects and facilitating the development of aquifer management plans.

## ACKNOWLEDGEMENTS

This paper is dedicated to the memory of Andrés Pérez-Estaún, whose friendship, skill, and dedication were essential to this work, but also for his continuous support, discussion and encouragement in the study of Barcelona Geology. The authors wish to acknowledge funding by the Spanish Ministry of Economy and Competitiveness projects: MEPONE (BIA2010-20244); MEDISTRAES (CGL2013-48869-C2- 1-R) and by the Generalitat de Catalunya (Grup Consolidat de Recerca: Grup d'Hidrologia Subterrània, 2009-SGR-1057). We also would like to acknowledge ADIF, GISA, DPTOP Generalitat de
Catalunya (Infrastructure administration), Ajuntament de Barcelona, Ajuntament de Sant Adrià del Besòs, Ajuntament de Badalona, Ajuntament de Santa Coloma de Gramenet, BCASA, BR Urban Development Agency; AMB Metropolitan Area of Barcelona (Local administrations), Catalan Water Agency (ACA) and several companies as UTELinia9, SACYR, ACCIONA, OHL, GEOCONSULT, FCC, Ferrovial-Agroman, INTECSA-INARSA, COPISA, COPCISA, among others, for their help in providing technical documents and data used for this study, and also, for their support throughout the
hydrogeological monitoring and assessment of different civil works.




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





**FIGURE CAPTIONS**

5      **FIGURE 1.** Orthophotograph of the northeastern Mediterranean coast of Spain covering the extent of the Barcelona city study area (IGN PNOA). Note the intense urbanization of this region, hindering outcrop existence. Coordinates are in Universal Transverse Mercator (UTM), Zone 31.

**FIGURE 2**. Outline of the methodology proposed for the efficient management of geological and hydrogeological data in an urban environment. The symbol of the folder and hammer refers to the process steps where the administration must intervene.

**FIGURE 3.** Geological map of the pre-quaternary substrate of Barcelona City and surroundings. This geological map was generated considering previous and additional observations performed for this work (see Section 3). See characteristics of the different geological units distinguished in Sections 4.1 and 4.2.

15    **FIGURE 4.** Geological map of Barcelona City and surroundings. This geological map was generated considering previous and additional observations performed for this work (see Section 3). See characteristics of the different geological units distinguished in Sections 4.1; 4.2 and 4.3.

**FIGURE 5**. Sketch of geological Cross Sections 4 and 7. Bottom: view plant of the cross sections position. These cross-sections were 20   performed using HEROS tools (Velasco et al., 2012a).

**FIGURE 6.** 3D model of the study area, showing the geometry of the different geological units distinguished. The right upper insert shows details of the 3D model in the norther area of the Besòs Delta. This model was created following the methodology proposed in this paper.

25



**FIGURE 1**

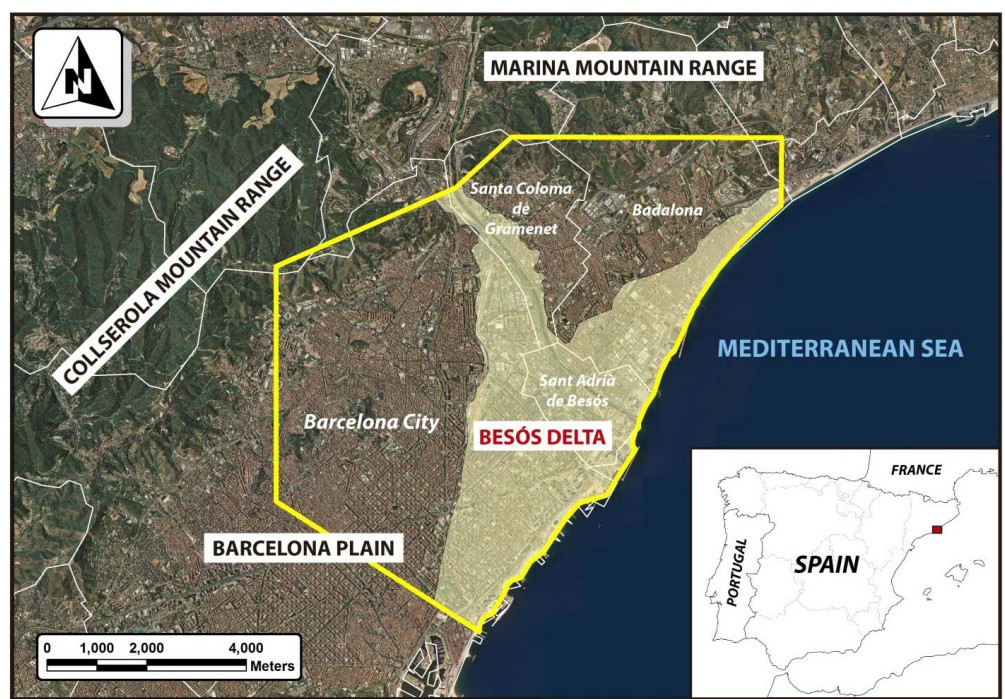

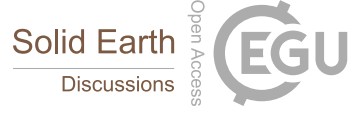

**FIGURE 2**

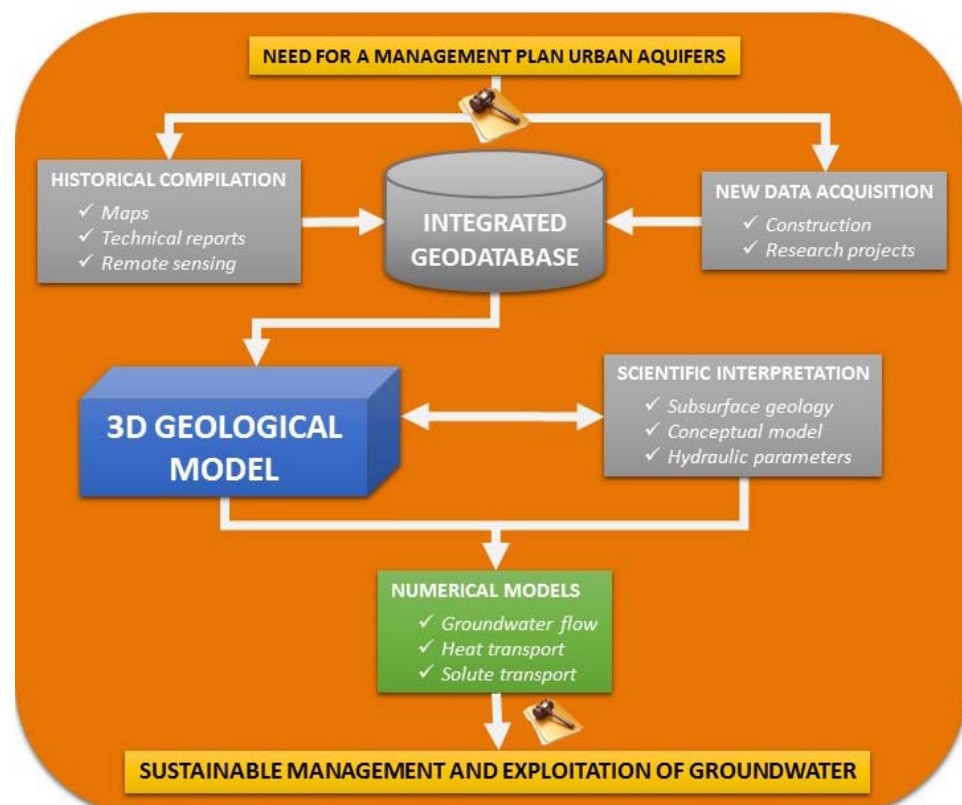



**FIGURE 3**

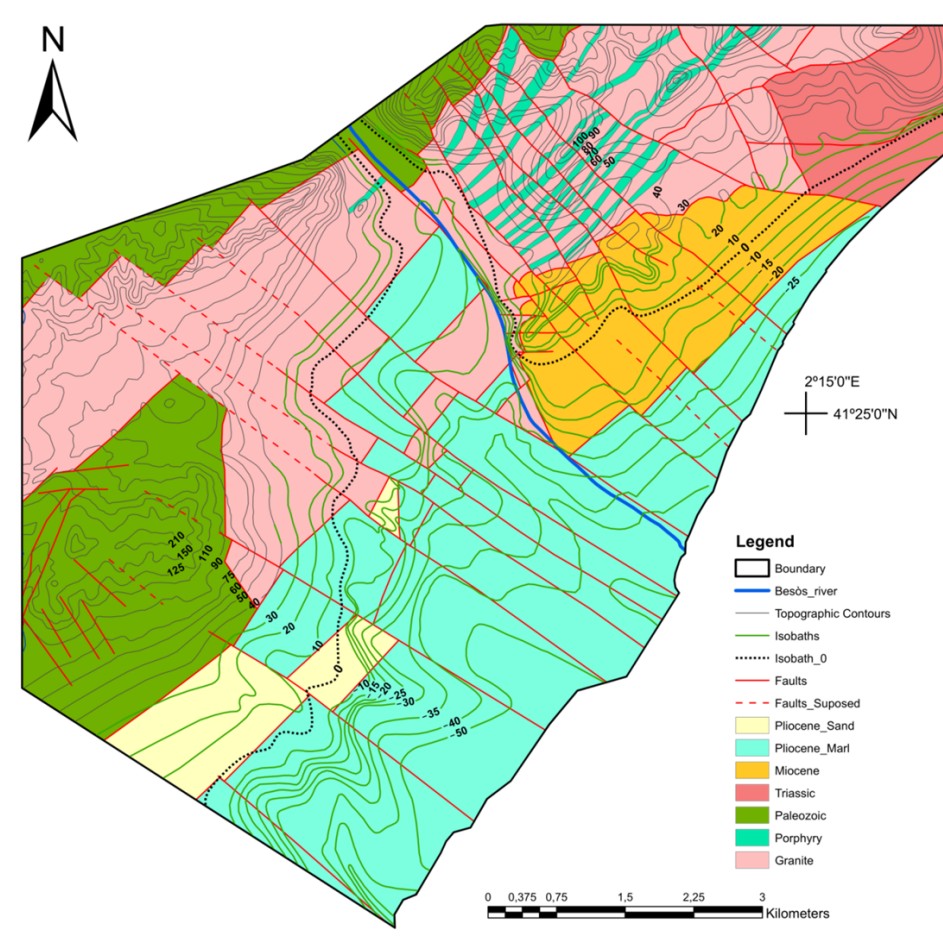



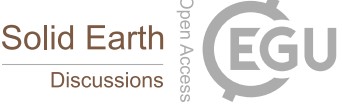

**FIGURE 4**

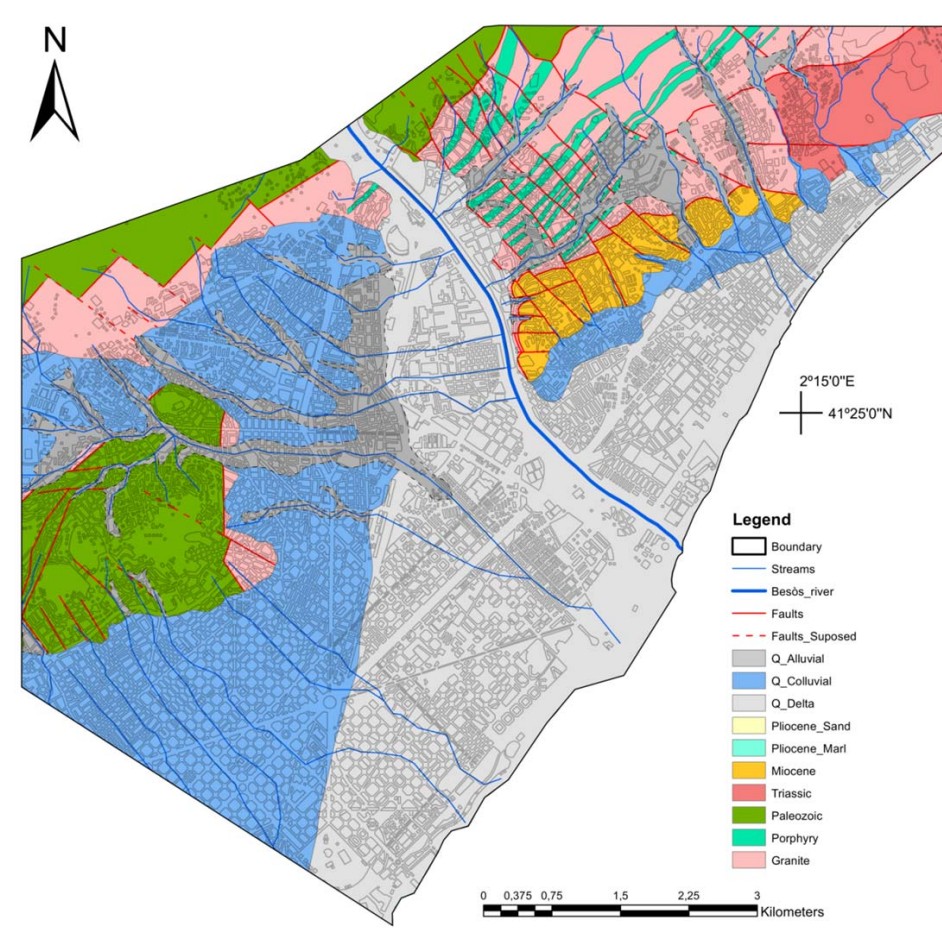



**FIGURE 5**

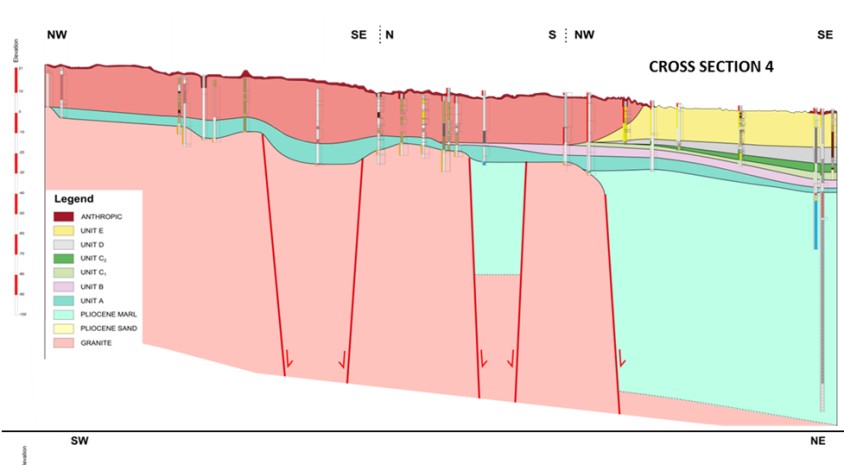

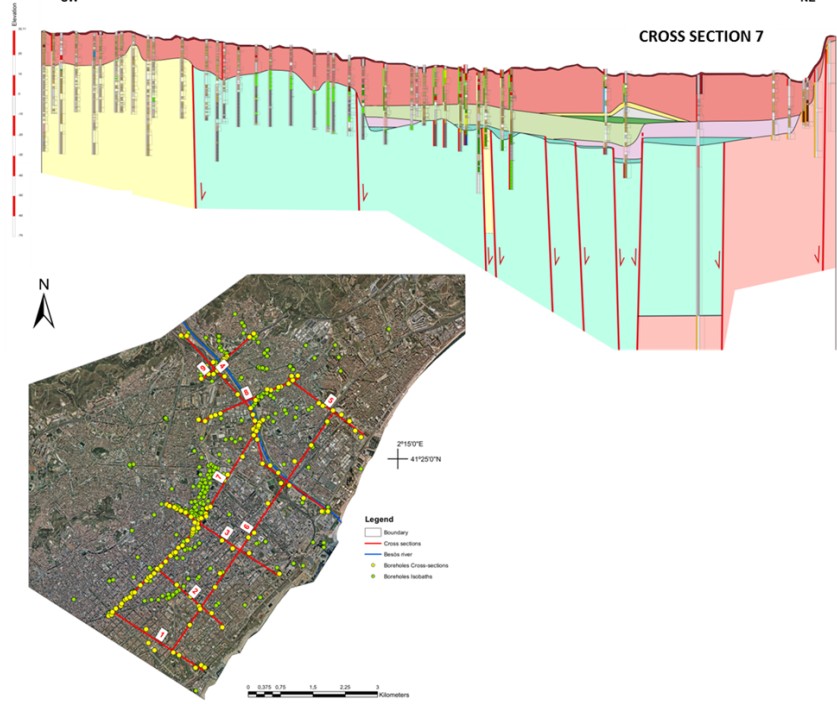





5    **FIGURE 6:**

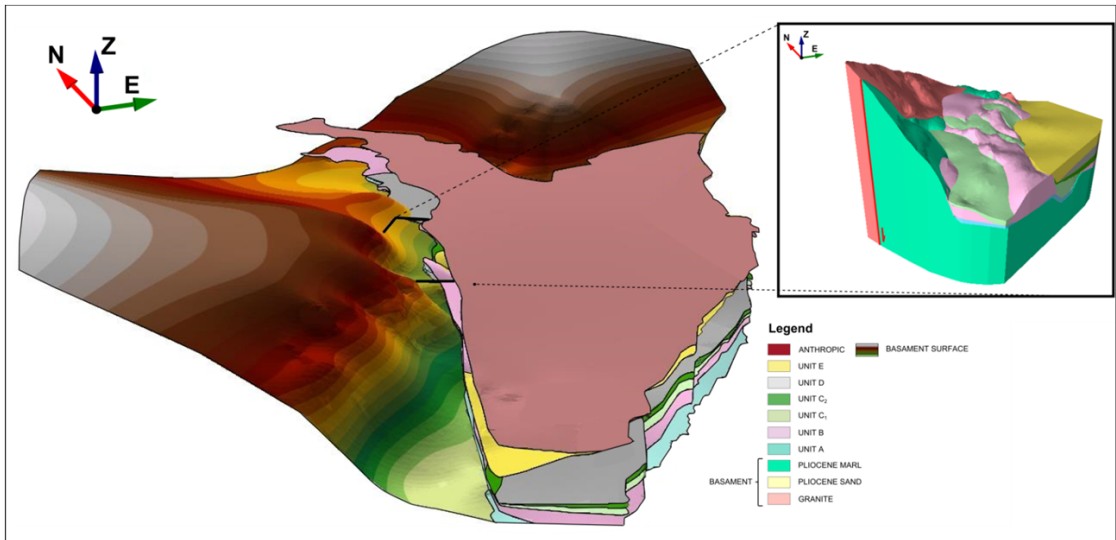