# Peer review of "A geological model for the management of subsurface data in the urban environment of Barcelona city."

_Solid Earth, 2016_

## Referee Comment (RC1) · Anonymous Referee #1 · 3 Jun 2016

[Figure]

[Figure]

[Figure]

[Figure]

[Figure]

**FIGURE 1**

[Figure]

[Figure]

[Figure]

[Figure]

**FIGURE 2**

[Figure]

[Figure]

[Figure]

**FIGURE 3**

[Figure]

[Figure]

**FIGURE 4**

[Figure]

[Figure]

[Figure]

**FIGURE 5**

[Figure]

[Figure]

[Figure]

**FIGURE 6:**

[Figure]

[referee-annotated manuscript omitted]

---

## Referee Comment (RC2) · Anonymous Referee #2 · 1 Jul 2016

General Comments

The article deals with an interesting issue as the geological knowledge in urban areas and its influence in groundwater resources management. Now a days, this is an important problems worldwide and novel approaches need to be found. The proposed methodology and the key point of including the administrative bodies highlight the advances proposed for this article.

However, the article does not follow a clear structure and seeming some time that your are reading another article when you change the section. In order to improve its impact, there are some major and minor points that need to be discussed/improved before its publication. The most important issues are the following:

[Figure]

- It is necessary an interrelation between the different sections of the article, being the most important linking properly the discussion with the previous sections. This is particularly relevant for the discussion.

- An improvement of the state of the art is needed as there many papers dealing with urban geology not mentioned in this article. It is necessary emphasize what is improving this study compared with many available.

- It is necessary to clarify the objectives and follow them along the text. From my point fo view there are three clear key points: 1) integrate the information to construct a robust geological model in a urban area and 2) use this information to improve aquifer management and 3) show the importance of collaborating with the government/administration. However, along the text some sections refer more to point 1) and others to 2) and 3). The explanation should be clearer in this way.

- The proposed model is based in many reports of previous studies and the administration. It is necessary to explain in more detail what is the information contained in these reports and how it has been integrated into the geological model.

Other minor issues that should be addressed are the following:

Line 22: What you mean for logistic difficulties? Scattered information? Little surface information/exploration available? Explain clearly to emphasize the importance of your work. Line 23-25: This part of the abstract should be modified as groundwater is not an example but the main application of the proposed work. Something like "The most important application of the proposed methodology has been applied to..." would be more realistic.

Line 15: explain in more detail what are the different solutions and ideas proposed in the literature. A quick search in scopus with the terms "groundwater", "management"

and "urban areas" gave me more 1200 articles. I think that a better "state of the art" can be done. Line 31: According the structure of the introduction and the importance that author give to groundwater management, I would divide the objective in three points as follows: (1) to develop a method for integrating geologic, hydrogeological, and geophysical data from an urban environment into a standardized, accessible database using effective data management tools and (2) to demonstrate how this challenge was met in the city of Barcelona (3) to show how the geology was modelled helping to improve aquifer management.

Line 19-22. Could you explain what means MAGNA, IGME and ICGC.

Line 1-4: The information included in the 714+1463 survey should be explained in more detail as the authors do in "Previous data and historical studies". It is necessary explain the kind of document compiled (i.e. technical report, year-book, etc.) and the geological information contained in these works (borehole description, maps, outcrops, etc). If the authors do not explain the type of data they are integrating in the proposed database, the proposed methodology can not be applied in other areas and the significance of the proposed work is completely not clear. Line 8: A few sentences should be added here explaining the basics about what is shown in Figure 2.

Section 4, page 4-7

The geology of the study site is very well explained with a lot of detail. However, this information is not used in the discussion. Then, I am not clear about what is the objective of adding this very well detailed geology.

Discussion: as mention before, Discussion section should be rewritten following the objectives of the paper. Line 10: Discussion should be point 5. Line 11-15: This

information is repeating what has been explained before.

Line 11-19: Despite there are different papers explaining the different tools attached to the proposed model/software, a more detailed explanation of each tool should be included. This information would allow the readers understand the potential of the proposed model/software.

Page 10.

Line 1-2: This sentence does not highlight the most important issue and/or most relevant information of the presented paper.

---

## Author Comment (AC1) · 29 Jul 2016

RESPONSE TO REFEREE #1 COMMENTS

We respond below to the comments made by the Referee. To facilitate reading we have pasted the original comments ("REFEREE #1") and our "RESPONSE".

REFEREE #1: The work presented is completely relevant for the management of water resources, and of high interest for the scientific community. The manuscript is synthetic and well written, with a complete list of references.

RESPONSE: We would like to begin by thanking the referee for his/her efforts, and we are very pleased for the overall positive assessment.

[Figure]

REFEREE #1: recomment to consider the following general observations to improve the manuscript, as well as the specific comments and technical corrections, specified all with comment notes in the attached pdf file:

RESPONSE: We have addressed most of the specific comments and technical corrections in the manuscript following referee's suggestions.

REFEREE #1: a) To complete Chapter 3 by explaining the general workflow presented in Figure 2 and how the model was built.

RESPONSE: We have rewritten some parts of this chapter in order to better explain the general work flow and to present a more detailed methodology for the model construction.

REFEREE #1: b) To include a geological map of the study area at a more regional scale, including geological elements commented in the text.

RESPONSE: We include a new general geological map of the study area.

REFEREE #1: c) Improve the figures (including scales, enlarging some of them, reviewing the legends, etc.).

RESPONSE: We have improved the majority of the figures following referee's suggestions.

---

## Author Comment (AC2) · 29 Jul 2016

RESPONSE TO REFEREE #2 COMMENTS We respond below to the comments made by the Referee. To facilitate reading we have pasted the original comments ("REFEREE #2") and our "RESPONSE".

REFEREE #2: General Comments The article deals with an interesting issue as the geological knowledge in urban areas and its influence in groundwater resources management. Now a days, this is an important problems worldwide and novel approaches need to be found. The proposed methodology and the key point of including the administrative bodies highlight the advances proposed for this article.

[Figure]

RESPONSE: We would like to begin by thanking the referee for his/her efforts, and we are very pleased for the overall positive assessment.

REFEREE #2: However, the article does not follow a clear structure and seeming some time that you are reading another article when you change the section. In order to improve its impact, there are some major and minor points that need to be discussed/improved before its publication. The most important issues are the following:

REFEREE #2: a) It is necessary an interrelation between the different sections of the article, being the most important linking properly the discussion with the previous sections. This is particularly relevant for the discussion.

RESPONSE: We agree with the referee's comment. For a better link between the previous sections and the discussion, we have rewritten some parts of the article (Introduction (chapter 1), Materials and methods (chapter 3), geological model (chapter 4) and the discussion (chapter 4). We believe that this revised version has substantially improved the original manuscript.

REFEREE #2: b) An improvement of the state of the art is needed as there many papers dealing with urban geology not mentioned in this article. It is necessary emphasize what is improving this study compared with many available.

RESPONSE: Thanks. As the referee suggests, there are additional works that are interesting and relevant to be included in the manuscript. We have considered and referenced them in the manuscript.

REFEREE #2: c) It is necessary to clarify the objectives and follow them along the text. From my point for view there are three clear key points: 1) integrate the information to construct a robust geological model in a urban area and 2) use this information to improve aquifer management and 3) show the importance of collaborating with the government/administration. However, along the text some sections refer more to point 1) and others to 2) and 3). The explanation should be clearer in this way.

RESPONSE: As explained before we have rewritten some sections of the paper. This includes a better statement of the objectives and their development along the text.

REFEREE #2: d) The proposed model is based in many reports of previous studies and the administration. It is necessary to explain in more detail what is the information contained in these reports and how it has been integrated into the geological model.

RESPONSE: We agree. We have included more information about the available surveys and data, and also about the methodology and the tools used to integrate that data into the geological model. (Item: 3.2. New data acquisition and methodology)

REFEREE #2: e) Other minor issues that should be addressed.

RESPONSE: We have addressed most of the specific comments and technical corrections in the manuscript following referee's suggestions.

---

## Author Comment (AC3) · 29 Jul 2016

Dear Editor,

We would like to begin by thanking you and the two reviewers for all of your efforts with this manuscript. The comments have been constructive and we have incorporated suggestions made by the Referees. As a result of these revisions, we believe that this revised version has substantially improved the original manuscript. Please find detailed responses to each of the Referee comments below.

Yours sincerely,

Enric Vázquez-Suñé and co-workers

Please also note the supplement to this comment:
http://www.solid-earth-discuss.net/se-2016-64/se-2016-64-AC3-supplement.pdf
* * *
[Figure]

**Supplement:**

Dear Sir,

We would like to begin by thanking you and the two Referees for all of your efforts with this manuscript. The comments have been constructive and we have incorporated suggestions made by the Referees. As a result of these revisions, we believe that this revised version has substantially improved the original manuscript. Please find detailed responses to each of the Referee comments below.

Yours sincerely,

Enric Vázquez-Suñé and co-workers

RESPONSE TO REVIEWER's COMMENTS

We respond below to the comments made by both Referees. To facilitate reading we have pasted the original comments in *italics* and our responses in blue standard font.

REFEREE #1

*The work presented is completely relevant for the management of water resources, and of high interest for the scientific community.*

*The manuscript is synthetic and well written, with a complete list of references.*

**We would like to begin by thanking the referee for his/her efforts, and we are very pleased for the overall positive assessment.**

*I recomment to consider the following general observations to improve the manuscript, as well as the specific comments and technical corrections, specified all with comment notes in the attached pdf file:*

**We have addressed most of the specific comments and technical corrections in the manuscript following referee's suggestions.**

*a) To complete Chapter 3 by explaining the general workflow presented in Figure 2 and how the model was built.*

**We have rewritten some parts of this chapter in order to better explain the general work flow and to present a more detailed methodology for the model construction.**

b) To include a geological map of the study area at a more regional scale, including geological elements commented in the text.

**We include a new general geological map of the study area.**

*c) Improve the figures (including scales, enlarging some of them, reviewing the legends, etc.).*

**We have improved the majority of the figures following referee's suggestions.**

**REFEREE #2**

*General Comments*
*The article deals with an interesting issue as the geological knowledge in urban areas and its influence in groundwater resources management. Now a days, this is an important problems worldwide and novel approaches need to be found. The proposed methodology and the key point of including the administrative bodies highlight the advances proposed for this article.*

**We would like to begin by thanking the referee for his/her efforts, and we are very pleased for the overall positive assessment.**

*However, the article does not follow a clear structure and seeming some time that your are reading another article when you change the section. In order to improve its impact, there are some major and minor points that need to be discussed/improved before its publication. The most important issues are the following:*

   *a) It is necessary an interrelation between the different sections of the article, being the most important linking properly the discussion with the previous sections. This is particularly relevant for the discussion.*

**We agree with the referee's comment. For a better link between the previous sections and the discussion, we have rewritten some parts of the article (Introduction (chapter 1), Materials and methods (chapter 3), geological model (chapter 4) and the discussion (chapter 4). We believe that this revised version has substantially improved the original manuscript.**

   *b) An improvement of the state of the art is needed as there many papers dealing with urban geology not mentioned in this article. It is necessary emphasize what is improving this study compared with many available.*

**Thanks. As the referee suggests, there are additional works that are interesting and relevant to be included in the manuscript. We have considered and referenced them in the manuscript.**

   *c) It is necessary to clarify the objectives and follow them along the text. From my point for view there are three clear key points: 1) integrate the information to construct a robust geological model in a urban area and 2) use this information to improve aquifer management and 3) show the importance of collaborating with the government/administration. However, along the text some sections refer more to point 1) and others to 2) and 3). The explanation should be clearer in this way.*

**As explained before we have rewritten some sections of the paper. This includes a better statement of the objectives and their development along the text.**

d) *The proposed model is based in many reports of previous studies and the administration. It is necessary to explain in more detail what is the information contained in these reports and how it has been integrated into the geological model.*

**We agree. We have included more information about the available surveys and data, and also about the methodology and the tools used to integrate that data into the geological model. (Item: 3.2. New data acquisition and methodology)**

e) Other minor issues that should be addressed.

**We have addressed most of the specific comments and technical corrections in the manuscript following referee's suggestions.**

---

## Author Comment (AC4) · 29 Jul 2016

[revised manuscript text omitted]

20  The development of comprehensive numerical models requires detailed geological models that represent the heterogeneity of the media in the three dimensions of the space.

In the absence of an extensive geological knowledge of the subsurface, which is also crucial for construction planning  and for security reasons (Huggenberger et al., 2013; Santanach et al., 2011; Velasco et al., 2012)

25  , the geological model as a basis for the hydrogeological models will be incomplete or will fail.

A sound geological analysis can be performed only if sufficient data are available. However, geological features are increasingly difficult to observe due to growing urbanization (Huggenberger and Epting, 2011), and the difficulty of performing geological and geophysical exploration in dense urban areas leads to a need for specific exploratory techniques.

In cities, it is very important to compile existing historical information. Although a large amount of information is typically

30  available (Culshaw and Price, 2011), much of this information is not stored in standardized or ordered formats but rather is dispersed among the various companies and public institutions that generated them using different protocols and criteria. Close collaboration is needed between an administration supplying the data and a research team investigating the geology (de Rienzo et al., 2008; Di Salvo et al., 2012). New methods of managing geological data in

urban areas are necessary: without these, it is much more complex and difficult to develop management plans for aquifers and subterranean construction in densely populated environments.

Thus,Apart from a detailed geological model, a reliable hydrogeological model must use other information available such as data from hydrometeorology, geography, hydrochemestry, etc. The data from each field complements the objectivesinterpretation of data from the rest of the fields. For example, a reliable geological analysis enables us to perform a proper parameterization of the study area that can be complemented with an aquifer test, or a detailed hydrochemical analysis allows us to re-interpret the geology (Velasco, 2013).

The aim of this study are (1) to develop a methodis twofold: First we present a methodology for integrating geologicall the required data (e.g. geological, hydrogeological, and geophysical data from an urban environment). The data that is from different sources (e.g. public administration or private companies) and presented in different format is converted into a standardized, accessible database using effective data-management tools and (2) to. These instruments facilitate the analysis, interpretation, pre and post processing of the data that is later used for modelling. Secondly, we show how this challenge was met in the city of Barcelona, i.e., how the geology was modelled and consequently how a better management of the aquifer was achieved. It should be highlighted that the workflow presented can be applied to other study areas.

**2. REGIONAL SETTING**

[revised manuscript text omitted]

The map of Barcelona based on the American flight of 1956 at  1:50,000 scale is highly useful for mapping  ancient geological exposures. In the middle of the 20th century, the first study of the Quaternary deposits was performed (Solé-Sabarís, 1963).

Geological mapping  from the second  series of the Spanish Geological Survey mapping plan at a 1:50,000 scale was used (Alonso et al., 1977). This work was later integrated with the current Institut Cartogràfic i Geològic de Catalunya (ICGC, 2005) and with the 1:25,000 scale geological map by Ventayol et al. (1978), Ventayol (2016). Sanz Parera (1988) compiled and synthesized geological and hydrogeological information regarding the Barcelona Plain; this information was updated by Riba and Colombo (2009).

**3.2. New data acquisition and methodology**

The absence of outcrops and the difficulty of applying certain techniques often used in geological exploration have meant adapting these techniques or, in many cases, developing new techniques to achieve the objectives.

During the previous 20 years, the Hydrogeology Group (CSIC-UPC) has performed extensive hydrogeological research in the vicinity of Barcelona, prompted mostly by the local governments of Barcelona, Santa Coloma, Badalona, etc. These studies focused on the management of groundwater resources and on the hydrogeological aspects of excavations, tunnels (e.g., the Metro and High Speed Train), drainages, and other infrastructures. These studies generated a large amount of geological and hydrogeological information that improved our knowledge of the subsurface below Barcelona.

The major source of information for this study consisted of exploratory surveys. First, a review of all existing surveys was performed, and all that were relevant were reinterpreted. Second, newer surveys that were part of various projects in which the research group was involved were incorporated. This review included a total of 714 surveys currently available in the study area, and a toal of 1462 surveys including those from the immediate vicinity. Furthermore, additional observations consisted of detailed studies of the limited outcrops available in the city parks and the road cuts as well as geological interpretation of old aerial photographs (taken in the 1950s just before urban sprawl affected the area).

All the information from construction projects, surveys, and nearby exposures together with geophysical profiles (Martí et al., 2008; Gámez et al., 2009; Velasco, 2013; Velasco et al., 2012a and 2012b) were standardized and entered into the geodatabase for modelling and analysis in a Geographic Information System (GIS) environment; concretely in ArcGIS (ESRI). This geospatial database is part of a complete on-going software platform that arrange all the available data into a coherent structure and provide support for their proper management, analysis and interpretation. The GIS-based tools provide mainly three families of utilities aimed at facilitating:

a) Geological analysis and 3D modelling using the set of tools HEROS and HEROS 3D. This instruments enable the visualization of stratigraphics columns, the generation of cross-sections and the generation of 3D models. Further information of these tools can be found in Velasco et al., 2012b and Alcaraz, 2016.

b) Hydrochemical analysis by using a set of tools QUIMET that perform quality controls, computation methods, statistical analysis and traditional graphical analysis techniques (e.g. Piper, Stiff, etc.). Further information in Velasco et al., (2014)

c) Hydrogeological data analysis and interpretation using HYYH tools, which are dedicated to query, represent, and analyse other hydrogeological data such as groundwater level, aquifer tests and well extractions or injections. Further details in Criollo et al., 2016.

All this enabled us to build a 3D geological model following the methodological approach represented in Figure 2. Some of these results are described in the following sections.

**4. GEOLOGICAL MODEL**

The geological model presented in this paper, include includes the entire stratigraphic succession rankingranging from Palaeozoic materialsrocks at the base, to recent deposits at the present top surface.

The Quaternary deposits unconformably overlie a heterogeneous substrate of various ages. First,that consist of Paleozoic, Triassic, Miocene, and Pliocene rocks. In this section we present first a synthesis of the lithologiclithological and structural characteristics of the pre-Quaternary substrate are described (Figures 3); these consist of Paleozoic, Triassic, Miocene, and Pliocene materials.). Later, the Quaternary stratigraphy of the BesósBesòs River delta is described (Figures 4 and 5).

**4.1. Paleozoic rocks and structural evolution**

The Paleozoic substrate consists of granitic and metamorphic rocks (Figure 3). The granitic rocks belong to the Paleozoic batholith that forms the core of the Catalan Coastal Range and the basement of the coastal shelf. They are plutonic rocks of granodioritic composition emplaced during the Carboniferous/Permian (Solé et al., 2002). During the latest stages of emplacement a large networks of porphyry dikes (locally granodioritic to granitic) and aplites, pegmatites, and porphyritic leucogranites of centimeter to meter scale were intruded. Physicochemical alteration of the granodiorites along joints resulted in scattered sandy material with variable amounts of clays and iron oxyhydroxides, locally referred to as the lehm. It is also often present in caliche horizons associated with major alteration zones, particularly in the lowlying areas, where the granites are more mafic.

The Paleozoic metamorphic rocks, are include Ordovician shales, schists, and quartzites, and Devonian limestones. Their northwestern margin is definedThey are bound in the north west by a normal fault. Two distinct, laterally extensive series can be distinguished. These rocks were intensely folded during the Hercynian orogeny and were affected by contact metamorphism associated to the intrusion by theof a granitic batholith in the form of a distinct zone of contact metamorphism.. This metamorphism that decreases with distance from the intrusion and createdis evidenced by three primary types of metamorphic rocks: hornfels, peliticsthat are pelitic 
[revised manuscript text omitted]

**FIGURE 1**

[Figure]

[Figure]

**FIGURE 2**

[Figure]

**FIGURE 3**

[Figure]

[Figure]

**FIGURE 4**

[Figure]

[Figure]

[Figure]

[Figure]

**FIGURE 6:**

[Figure]

---

## Author Comment (AC5) · 29 Jul 2016

**A geological model for the management of subsurface data in the urban environment of Barcelona city and surrounding area.**

Enric Vázquez-Suñé[1,2], Miguel Ángel Marazuela[1,2,3], Violeta Velasco[1,2], Marc Diviu[1,2], Andrés Pérez-Estaún[4], Joaquina Álvarez-Marrón[4]

[1] Institute of Environmental Assessment and Water Research IDÆA - CSIC. Jordi Girona 18, 08034 Barcelona, Spain
[2] Associated Unit: Hydrogeology Group (UPC-CSIC)
[3] Department of Geotechnical Engineering and Geosciences, Universitat Politècnica de Catalunya (UPC), Jordi Girona 1-3, 08034 Barcelona, Spain
[4] Institute of Earth Sciences Jaume Almera ICTJA - CSIC, Lluís Solé Sabarís s/n, 08028 Barcelona, Spain

*Correspondence to*: Enric Vázquez-Suñé  (enric.vazquez@idaea.csic.es)

**Abstract.** The overdevelopment of cities since the industrial revolution has shown the need to incorporate a sound geological knowledge in the management of required subsurface infrastructures and in the assessment of increasingly needed groundwater resources. Also, the scarcity of outcrops and the technical difficulty to conduct underground exploration in urban areas highlights the importance of implementing efficient management plans that deal with the legacy of heterogeneous subsurface information. To deal with these difficulties, a methodology has been proposed to integrate all the available spatiotemporal data into a comprehensive spatial database and a set of tools that facilitates the analysis and processing of the existing and newly added data for the city of Barcelona (NE Spain). Here we present the resulting actual subsurface 3D geological model that incorporates and articulates all the information stored in the database. The methodology applied to Barcelona city benefited from a good collaboration between administrative bodies and researchers that enabled the realization of a comprehensive geological database despite logistic difficulties. Currently, the public administration and also private sectors, both benefit from the geological understanding acquired in the Barcelona city. For example, when preparing the hydrogeological models used in groundwater assessment plans. The methodology further facilitates the continuous incorporation of new data for the implementation and sustainable management of urban groundwater and also contributes to significantly reduce the costs of new infrastructures.

**1. INTRODUCTION**

The need for a groundwater supply has led people to settle near natural water sources since ancient times, usually near major rivers. For this reason, many of the world's major cities are located above alluvial or deltaic aquifers (Carlson et al., 2011; Davies, 2015; Gleeson et al., 2015; Tessler et al., 2015). Currently, particularly with the industrialization that began in the 1950s, most of these aquifers have been severely affected in terms of the quantity and quality of their waters (Aeschbach-Hertig and Gleeson, 2012; Gleeson et al., 2012b).

Urban expansion generates an enormous demand for infrastructure and, paradoxically, progressively less space for construction (Grimm et al., 2008). As a result, cities must grow downward, which requires subterranean construction (e.g., transportation networks and underground parking garages). This subterranean infrastructures usually interact with shallow aquifers producing frequently negative repercussion on them (Attard et al., 2015; Ferguson, 2004; Ferguson and Woodbury, 2007; Kazemi, 2011; McDonald et al., 2014; Menberg et al., 2013; Taniguchi et al., 2008; Vázquez-Suñé et al., 2005; Vörösmarty et al., 2000). Various contamination pathways associated with this problem include spillways, sewage networks, and discharges into wells. Another result of urbanization is an abundance of hardscapes, which severely reduce the amount of aquifer recharge (Vázquez-Suñé et al., 2010).

The European Water Framework Directive was developed in response to these problems faced by society in the 21st century (WFD, 2000). Its objective was to unify water management in the European Union by defining long-term needs and actions to be performed in various countries. It is an important step in the management and assessment of groundwater in general. However, its regulations are typically too lax and do not provide for adequate management of urban aquifers required for the sustainable, controlled exploitation of water and geothermal resources (Epting et al., 2013; García-Gil et al., 2015b; Gleeson et al., 2012a; Vázquez-Suñé et al., 2006).

The literature contains several solutions and ideas for general groundwater management or for assessment of specific problems related to subsurface constructions in urban areas. (García-Gil et al., 2015c; Parriaux et al., 2004; Zhou et al., 2015). Almost all of these studies conclude that numerical models of current and future aquifer conditions are the primary tool for accomplishing this objective. The development of comprehensive numerical models requires detailed geological models that represent the heterogeneity of the media in the three dimensions of the space.

In the absence of an extensive geological knowledge of the subsurface, which is also crucial for construction planning and for security reasons (Huggenberger et al., 2013; Santanach et al., 2011; Velasco et al., 2012), the geological model as a basis for the hydrogeological models will be incomplete or will fail.

A sound geological analysis can be performed only if sufficient data are available. However, geological features are increasingly difficult to observe due to growing urbanization (Huggenberger and Epting, 2011), and the difficulty of performing geological and geophysical exploration in dense urban areas leads to a need for specific exploratory techniques.

In cities, it is very important to compile existing historical information. Although a large amount of information is typically available (Culshaw and Price, 2011), much of this information is not stored in standardized or ordered formats but rather is dispersed among the various companies and public institutions that generated them using different protocols and criteria. Close collaboration is needed between an administration supplying the data and a research team investigating the geology (de Rienzo et al., 2008; Di Salvo et al., 2012;). New methods of managing geological data in urban areas are necessary: without these, it is much more complex and difficult to develop management plans for aquifers and subterranean construction in densely populated environments.

Apart from a detailed geological model, a reliable hydrogeological model must use other information available such as data from hydrometeorology, geography, hydrochemestry, etc. The data from each field complements the interpretation of data

from the rest of the fields. For example, a reliable geological analysis enables us to perform a proper parameterization of the study area that can be complemented with an aquifer test, or a detailed hydrochemical analysis allows us to re-interpret the geology (Velasco, 2013).

The aim of this study is twofold: First we present a methodology for integrating all the required data (e.g. geological, hydrogeological, and geophysical data). The data that is from different sources (e.g. public administration or private companies) and presented in different format is converted into a standardized, accessible database using effective data-management tools. These instruments facilitate the analysis, interpretation, pre and post processing of the data that is later used for modelling. Secondly, we show how this challenge was met in the city of Barcelona, i.e., how the geology was modelled and consequently how a better management of the aquifer was achieved. It should be highlighted that the workflow presented can be applied to other study areas.

**2. REGIONAL SETTING**

[revised manuscript text omitted]

All the information from construction projects, surveys, and nearby exposures together with geophysical profiles (Martí et al., 2008; Gámez et al., 2009; Velasco, 2013; Velasco et al., 2012a and 2012b) were standardized and entered into the
20 geodatabase for modelling and analysis in a Geographic Information System (GIS) environment; concretely in ArcGIS (ESRI). This geospatial database is part of a complete on-going software platform that arrange all the available data into a coherent structure and provide support for their proper management, analysis and interpretation. The GIS-based tools provide mainly three families of utilities aimed at facilitating:

a) Geological analysis and 3D modelling using the set of tools HEROS and HEROS 3D. This instruments enable the
25 visualization of stratigraphics columns, the generation of cross-sections and the generation of 3D models. Further information of these tools can be found in Velasco et al., 2012b and Alcaraz, 2016.

b) Hydrochemical analysis by using a set of tools QUIMET that perform quality controls, computation methods, statistical analysis and traditional graphical analysis techniques (e.g. Piper, Stiff, etc.). Further information in Velasco et al., (2014)

30 c) Hydrogeological data analysis and interpretation using HYYH tools, which are dedicated to query, represent, and analyse other hydrogeological data such as groundwater level, aquifer tests and well extractions or injections. Further details in Criollo et al., 2016.

All this enabled us to build a 3D geological model following the methodological approach represented in Figure 2. Some of these results are described in the following sections.

**4. GEOLOGICAL MODEL**

The geological model presented in this paper includes the entire stratigraphic succession ranging from Palaeozoic rocks at the base to recent deposits at the surface.

The Quaternary deposits unconformably overlie a heterogeneous substrate that consist of Paleozoic, Triassic, Miocene, and Pliocene rocks. In this section we present first a synthesis of the lithological and structural characteristics of the pre-Quaternary substrate (Figures 3). Later, the Quaternary stratigraphy of the Besòs River delta is described (Figures 4 and 5).

**4.1. Paleozoic rocks and structural evolution**

The Paleozoic substrate consists of granitic and metamorphic rocks (Figure 3). The granitic rocks belong to the Paleozoic batholith that forms the core of the Catalan Coastal Range and the basement of the coastal shelf. They are plutonic rocks of granodioritic composition emplaced during the Carboniferous/Permian (Solé et al., 2002). During the latest stages of emplacement a large networks of porphyry dikes (locally granodioritic to granitic) and aplites, pegmatites, and porphyritic leucogranites of centimeter to meter scale were intruded. Physicochemical alteration of the granodiorites along joints resulted in scattered sandy material with variable amounts of clays and iron oxyhydroxides, locally referred to as the lehm. It is also often present in caliche horizons associated with major alteration zones, particularly in the lowlying areas, where the granites are more mafic.

The Paleozoic metamorphic rocks include Ordovician shales, schists and quartzites, and Devonian limestones. They are bound in the north west by a normal fault. Two distinct, laterally extensive series can be distinguished. These rocks were intensely folded during the Hercynian orogeny and were affected by contact metamorphism associated to the intrusion of a granitic batholith. This metamorphism that decreases with distance from the intrusion is evidenced by three primary types of metamorphic rocks: hornfels, that are pelitic 
[revised manuscript text omitted]

**FIGURE 1**

[Figure]

**FIGURE 2**

[Figure]

**FIGURE 3**

[Figure]

**FIGURE 4**

[Figure]

**FIGURE 5**

[Figure]

[Figure]

5 **FIGURE 6:**

[Figure]